# Rhodium hydride enabled enantioselective intermolecular C–H silylation to access acyclic stereogenic Si–H

Kun An[1,3], Wenpeng Ma[1,3], Li-Chuan Liu[1], Tao He[1], Guiyu Guan[1], Qing-Wei Zhang [2✉] & Wei He [1✉]

The tremendous success of stereogenic carbon compounds has never ceased to inspire researchers to explore the potentials of stereogenic silicon compounds. Intermolecular C–H silylation thus represents the most versatile and straightforward strategy to construct C–Si bonds, however, its enantioselective variant has been scarcely reported to date. Herein we report a protocol that allows for the enantioselective intermolecular C–H bond silylation, leading to the construction of a wide array of acyclic stereogenic Si–H compounds under simple and mild reaction conditions. Key to the success is (1) a substrate design that prevents the self-reaction of prochiral silane and (2) the employment of a more reactive rhodium hydride ([Rh]-H) catalyst as opposed to the commonly used rhodium chloride ([Rh]-Cl) catalyst. This work unveils opportunities in converting simple arenes into value-added stereogenic silicon compounds.

[1] MOE Key Laboratory of Bioorganic Phosphorus Chemistry & Chemical Biology and School of Pharmaceutical Sciences, Tsinghua University, Beijing, PR China. [2] Department of Chemistry, University of Science and Technology of China, Hefei, PR China. [3]These authors contributed equally: Kun An, Wenpeng Ma. ✉email: qingweiz@ustc.edu.cn; whe@mail.tsinghua.edu.cn

Catalytic functionalizations of C–H bonds including their asymmetric variants are at the center stage of modern chemistry[1–4]. Catalytic C–H silylations[5–10] thus represent the most straightforward strategy to access value-added silicon compounds from simple arenes or heteroarenes (Fig. 1a). Along this line, a plethora of catalyst systems including Lewis acids[11–16], bases[17–19], or transition metals[20–43] had been shown to affect intermolecular C–H silylations under various conditions. In particular, Hartwig and coworkers[28] have successfully developed a rhodium/bisphosphine catalyzed reaction between MeSi-H(OTMS)₂ and various arenes, which demonstrated a broad C–H bond scope and an excellent regioselectivity. Despite the enormous knowledge and important progresses, enantioselective intermolecular C–H silylation has been scarcely reported to date.

In sharp contrast, enantioselective intramolecular C–H silylation[44–50] has met with great successes (Fig. 1b). Takai, Kuminobu and coworkers[51] were the pioneers to demonstrate the asymmetric induction by a Rh/BINAP system in their intramolecular double C–H silylations. Building on this concept, a number of research groups including us have developed various Rh catalyzed enantioselective intramolecular C–H silylation reactions[52–60] to construct stereogenic silicons constrained in a ring (cyclic stereogenic silicons). In these intramolecular reactions, the C–H bonds were placed in well-designed positions such that they were predisposed to intercept the Si-[Rh] intermediate, thus overcoming the low reactivity and poor regioselectivity of the C–H bonds.

Since such a pre-organized reaction pattern enjoyed by the intramolecular CH silylation is not available in the intermolecular C–H silylation, the low reactivity of the C–H bonds brings up significant challenges. Because the C–H bonds are not readily available to react with the Si-[Rh] intermediate, the Si-[Rh] tends to react with itself, leading exclusively to Si–Si (dehydrogenative coupling)[61–63] or Si–C metathesis (silane redistribution)[64–66] side products. Such low stabilities of the Si–H precursors had been repeatedly documented in literature examples. For instance, in the

reports from the Grubbs and Stoltz groups[17] as well as the Hartwig group[28], excess amount of precursor silanes were used. Moreover, only a few of the existing C–H silylation reactions[12,14,15,38,43] had produced an intrinsically chiral silicon (i.e., a silicon bearing three different groups), suggesting that the development of enantioseletive C–H silylation was not merely to find a proper chiral catalyst system. Instead, it is clear that three prerequisites have to be met: (1) a prochiral silane that can be converted into a stereogenic Si in an enantioselective fashion; (2) a substrate design that circumvents the self-reaction of the prochiral silane, and (3) a catalyst system that has high reactivity towards the intermolecular C–H bonds, thus further alleviating the self-reaction of the prochiral silane.

Herein, we report an example of enantioselective intermolecular C–H silylation (Fig. 1c), which stems from our long-term research in silicon chemistry. First, we choose silacyclobutanes (SCBs) as the prochiral silanes based on our earlier findings[54,67]. Second, we employ [Rh]-H as the catalyst, since our combined computational and experimental study[68] have identified it as the true and more reactive catalyst for C–H silylation. Third, we uncover that a blocking group (but not a directing group) on the aryl substituent effectively prevents the oligomerization of SCBs. These considerations collectively lead to the current protocol, which allows the access to a wide array (77 examples) of acyclic stereogenic silanes in high yields and excellent enantioselectivities (up to 96% ee).

## Results and discussion

**Optimization of reaction conditions**. We commenced our study by using silacyclobutane **1a** and 2-methylthiophene **2a** as the model substrates. Since our earlier work[67] showed that the catalyst and solvent of choices were Rh(cod)Cl ([Rh]-Cl) and toluene, respectively, we investigated a panel of diphosphine ligands using this system first. Importantly, no reaction took place in the absence of a ligand (Table 1, entry 1), dispelling the

**a** Catalytic Intermolecular C-H Silylation

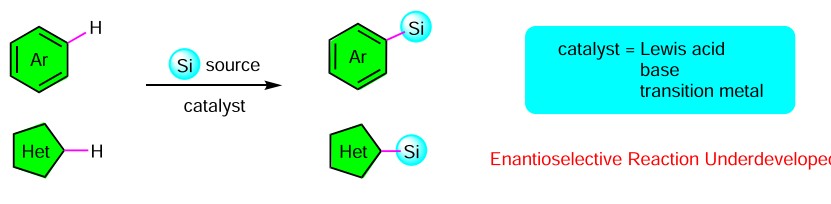

**b** Enantioselective Intramolecular C-H Silylation

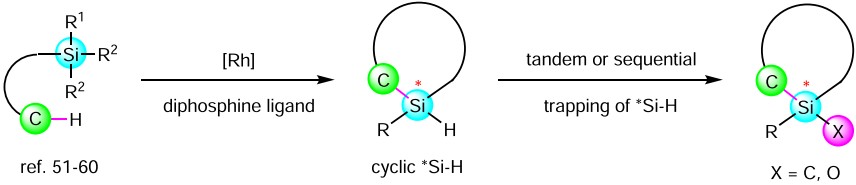

**c** This Work: Enantioselective Intermolecular C-H Silylation

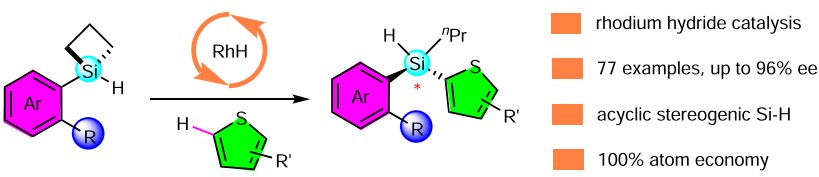

**Fig. 1 Direct C–H silylation for the synthesis of silicon-stereogenic silanes. a** Racemic catalytic intermolecular C–H silylation. **b** Enantioselective intramolecular C–H silylation (well established). **c** Enantioselective intermolecular C–H silylation (this work).

**Table 1 Optimization of reaction conditions[a,b].**

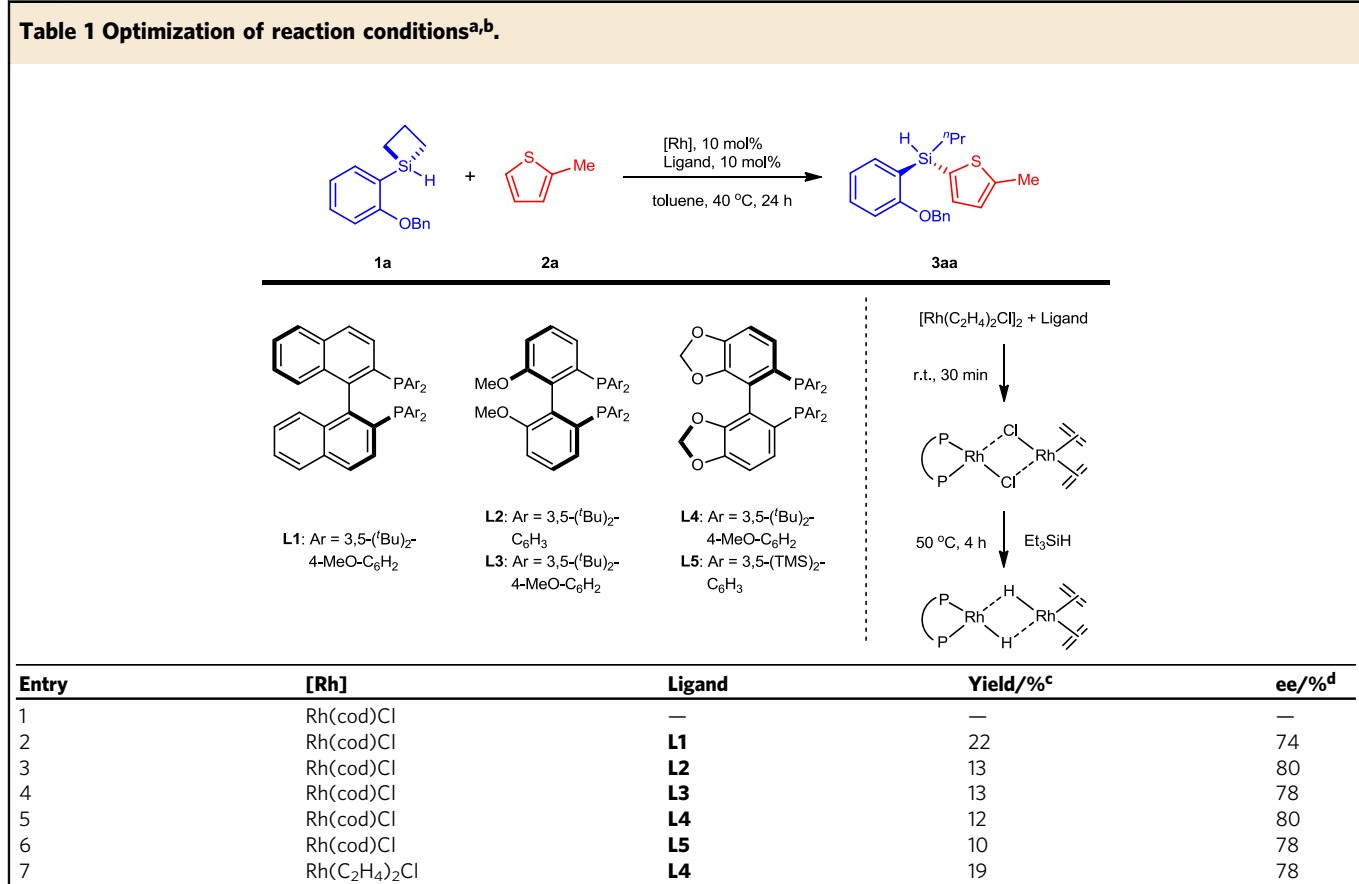

| Entry | [Rh] | Ligand | Yield/%[c] | ee/%[d] |
|---|---|---|---|---|
| 1 | Rh(cod)Cl | — | — | — |
| 2 | Rh(cod)Cl | **L1** | 22 | 74 |
| 3 | Rh(cod)Cl | **L2** | 13 | 80 |
| 4 | Rh(cod)Cl | **L3** | 13 | 78 |
| 5 | Rh(cod)Cl | **L4** | 12 | 80 |
| 6 | Rh(cod)Cl | **L5** | 10 | 78 |
| 7 | Rh(C$_2$H$_4$)$_2$Cl | **L4** | 19 | 78 |
| 8 | Rh(**L2**)-H | | 54 | 84 |
| 9 | Rh(**L3**)-H | | 48 | 88 |
| 10 | Rh(**L4**)-H | | 61 | 84 |
| 11 | Rh(**L5**)-H | | 68 | 76 |
| 12[e] | Rh(**L3**)-H | | 57(46)[f] | 92 |

[a]Reaction conditions: **1a** (0.1 mmol), **2a** (0.2 mmol), [Rh] catalyst (10 mol%), Ligand (10 mol%), toluene (1.0 mL) in a sealed Schlenk tube. [b]For the [Rh(**L**)-H] species, [Rh(C$_2$H$_4$)$_2$Cl]$_2$ (5 mol%), Ligand (5 mol%), Et$_3$SiH (0.5 equiv.) were employed. [c]Yield of isolated product. [d]Enantioexcesses determined by HPLC analysis on a chiral stationary phase. [e]The reaction was conducted at room temperature (r.t.) for 48 h. [f][Rh(C$_2$H$_4$)$_2$Cl]$_2$ (2.5 mol%), Ligand (2.5 mol%), Et$_3$SiH (0.25 equiv.) were employed.

concern about background reactions. When (S)-DTBM-BINAP (**L1**, entry 2) was employed, the desired tertiary hydrosilane **3aa** was obtained in a low yield (22%) with a moderate enantios-electivity (74% ee). The MeO-Biphep ligands **L2** and **L3** that have smaller dihedral angles exhibited a decreased reactivity yet an enhanced enantioselectivity, giving poorer yields (13%) and good enantioselectivities (80% and 78% ee, respectively). Segphos ligands **L4** and **L5** (entries 5 and 6) that worked well in our intramolecular C–H silylations[54,57,69] gave even poorer yields (12% and 10%, respectively). In order to improve the reaction, we tested the more reactive Rh precatalyst Rh(C$_2$H$_4$)$_2$Cl (entry 7). This catalyst showed a slightly improved yield (19%) and comparable enantioselectivity (78% ee). We performed in situ [1]H NMR monitoring on the reaction and observed that the substrate **1a** suffered rapid oligomerization while the thiophene remained intact. This observation was consistent with the prior findings[70] that the C–H bond partner was not capable of intercepting the [Rh]-Si intermediate before it was engaged in self-reaction.

We thus turned our attention to [Rh]-H catalyst based on our earlier findings that it was the true catalyst for the intramolecular C–H silylation of SCB with aryl C–H bonds[68]. To this end, we prepared a number of pre-formed [Rh]-H catalysts (Supplementary Figs. 1–8). Even at a lower catalyst loading (5 mol%) than the [Rh]-Cl catalyst (10 mol%), the [Rh]-H catalyzed reactions showed systemically improved yields (cf. entries 2–6 vs. entries 8–11). Among them, Rh(**L3**)-H gave the best enantioselectivity of

88% ee (entry 9). There was probably a tradeoff between the reaction yield and enantioselectivity. For example, the bulkiest ligand **L5** saw the highest yield (68%) but the lowest enantios-electivity (76% ee) (entry 11), while ligand **L3** gave a balanced performance in terms of yield and ee (entry 9). Importantly, the employment of [Rh]-H allowed the reaction to proceed at room temperature, albeit a longer reaction time (48 h) was required. Under this new set of conditions with Rh(**L3**)-H as the catalyst, the reaction showed a better yield (57%) and excellent enantioselectivity (92% ee) (entry 12).

**Substrate scope**. We then demonstrated the scope of C–H bond partners in the forms of thiophenes and benzothiopene under the optimized reaction conditions (Fig. 2). First, a number of 2-substituted thiophenes with different electronic and steric demands were tested, which all successfully produced the corresponding chiral silanes (**3ab**-**3ag**). Second, the unsubstituted thiophene reacted smoothly to give the desired product **3ah** in excellent enantioselectivty (90% ee). Third, alike the 2-substituted thiophenes, the 3-subsituted thiophenes showed good compatibility with the reaction, affording the products **3ai**-**3al** in good yields and high enantioselectivities (80–90% ee). It was seen that collectively the substitution on the 3-position had a less impact on the reaction outcomes than that on the 2-position which was probably attributed to the initial coordination of thiophenes to

**Fig. 2 Reactions with different C–H partners.** [a]Reaction conditions: **1a** (0.1 mmol), **2** (0.2 mmol), [Rh(C$_2$H$_4$)$_2$Cl]$_2$ (5 mol%), **L3** (5 mol%), Et$_3$SiH (0.5 equiv.), toluene (1.0 mL) in a sealed Schlenk tube. [b]Yield of isolated product. [c]Enantiomer ratio determined by HPLC analysis on a chiral stationary phase. [d]The reaction was conducted on 1.0 mmol scale. r.t. = room temperature.

the pre-formed [Rh]-H catalyst (Supplementary Fig. 14, 15). Finally, di-substituted thiophene and benzothiophene reacted smoothly to furnish the desired products **3am** and **3an** in excellent enantioselectivities (90% and 86% ee, respectively). However, 3,4-dimethylthiophene was not compatible with current transformation probably due to the steric effect of methyl group, and other heterocycles such as furan, pyrrole, pyridine, and indole appeared to be unsuccessful C–H partners in this reaction. We also attempted SCBs bearing unsubstituted phenyl (i.e., **1a** devoid of the –OBn group), but the reaction substrates suffered complete oligomerization and no silylation products were detected. This observation suggests that the substitution ortho- to the phenyl-Si bond is essential for the success of the reaction, which likely prevents the [Rh]-Si self-reaction by virtue of steric hindrance.

Next, we showed that a number of blocking groups instead of benzyl ethers were well tolerated (Fig. 3). For illustration, we employed three representative C–H partners, namely, 2-methylthiopene (**2a**), 3-methylthoiphene (**2i**) and benzothiophene (**2n**). Four alkyl ethers, methyl- (**3b**), ethyl- (**3c**), n-butyl- (**3d**) and isopropyl- (**3e**) were examined first. Across the panel these substrates showed consistent yields (50–69%) and high enantioselectivities (86–94% ee). It was observed that the **3e** accumulatively performed better than other ethers, probably due to its biggest steric hindrance derived from the isopropyl group. The insensitivity of the reaction outcomes to the ether substitution suggests that many

other types of ether substitution should be compatible with this reaction if desired. We also replaced the ethers with simple alkyls as seen in the cases of **1 f** and **1 g**. The desired products **3fa-3gn** were obtained in even higher yields (76–84%) and excellent enantioselectivities (90–95% ee). These two examples indicate that many alkyl substitutions ortho- to the Si should be tolerated by our method. Interestingly, ortho-phenyl substitution was also compatible if the available intramolecular C–H silylation sites were sterically shielded. As shown in the cases of **3 ha**, **3 hi** and **3 hn**, the desired products were obtained in ~60% yields and ~80% ee. Such a pattern was successfully extended to small heterocycle framework such as in the case of **1i**. Thus, acyclic stereogenic silicons **3ia**, **3ii** and **3in** with two heterocycle substituents were obtained in 72–74% yields with 90–93% ee. The different choices of the groups on the ortho-position indicate that they are blocking groups but not directing groups. Many other functionalities that are stable towards Si-H, such as F-, Cl-, CF$_3$-, carbocycles and so on, should be compatible with this reaction.

We further demonstrated that poly-substitutions on the aryl ring were tolerated (Fig. 4). For illustration purpose, we used ethyl-ether when appropriate. Again, 2-methylthiophene (**2a**), 3-methylthiophene (**2i**) and benzothiophene (**2n**) were used as the representative C–H partners. This matrix of 11×3 reactions (**3ja-3tn**) gave moderate to good yields and high enantioselectivities. (1) A number of tri-substituted phenyls (**1j-1r**) on the SCBs were explored. In this class of substrates, we first walked a

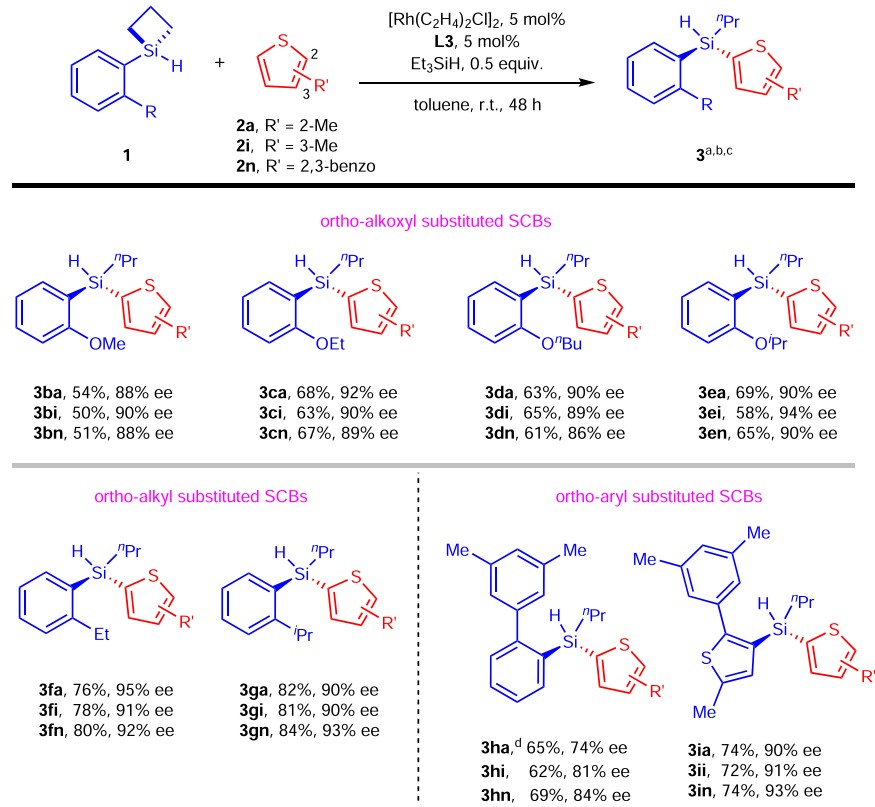

**Fig. 3 Scope of silacyclobutanes (SCBs) in terms of blocking groups.** [a]Reaction conditions: **1** (0.1 mmol), **2** (0.2 mmol), [Rh(C₂H₄)₂Cl]₂ (5 mol%), **L3** (5 mol%), Et₃SiH (0.5 equiv.), toluene (1.0 mL) in a sealed Schlenk tube. [b]Yield of isolated product. [c]Enantiomer ratio determined by HPLC analysis on a chiral stationary phase. [d]ee of **3 ha** was determined by its silanol derivative. r.t. = room temperature.

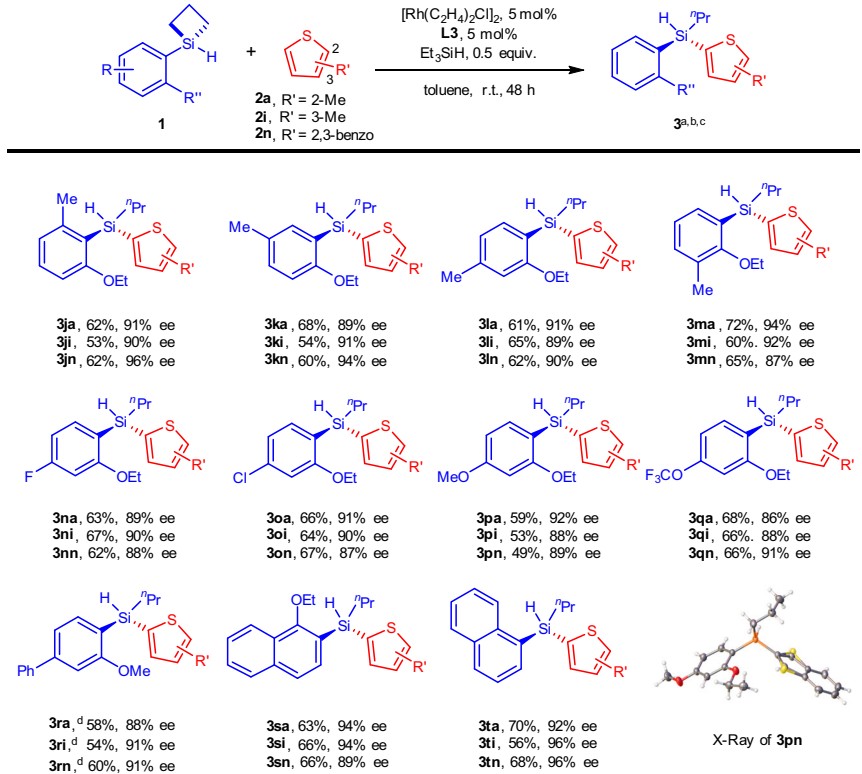

**Fig. 4 Substrate scope on the aryl ring.** [a]Reaction conditions: **1** (0.1 mmol), **2** (0.2 mmol), [Rh(C₂H₄)₂Cl]₂ (5 mol%), **L3** (5 mol%), Et₃SiH (0.5 equiv.), toluene (1.0 mL) in a sealed Schlenk tube. [b]Yield of isolated product. [c]Enantiomer ratio determined by HPLC analysis on a chiral stationary phase. [d][Rh(C₂H₄)₂Cl]₂ (10 mol%), **L3** (10 mol%), toluene (1.0 mL) were employed. r.t. = room temperature.

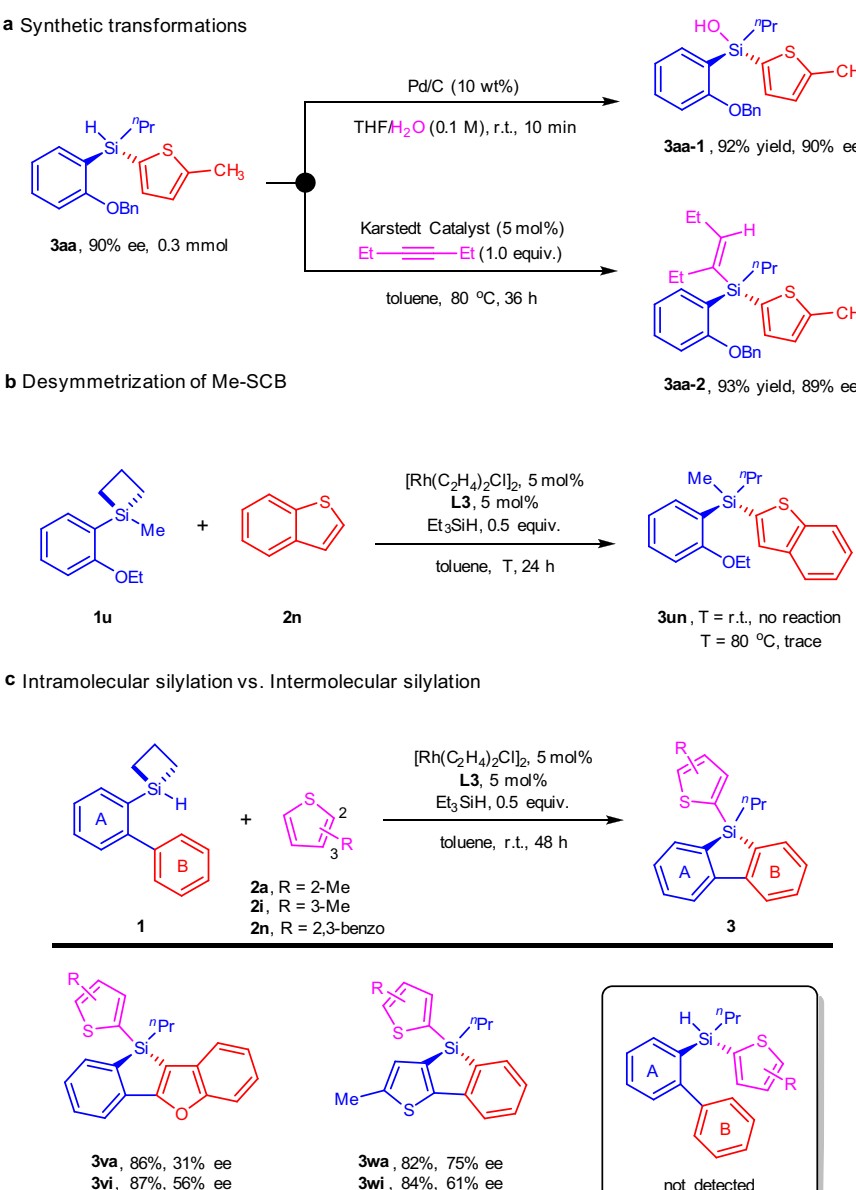

**Fig. 5 Preliminary elaborations and control experiments. a** Synthetic transformations of monohydrosilane **3aa**. **b** Desymmetrization of methyl-substituted silacyclobutane (SCB). Reaction conditions: **1u** (0.1 mmol), **2n** (0.1 mmol), [Rh(C$_2$H$_4$)$_2$Cl]$_2$ (5 mol%), **L3** (5 mol%), Et$_3$SiH (0.5 equiv.), toluene (1.0 mL) in a sealed Schlenk tube. **3un** was detected by GC-MS analysis. T = temperature. **c** Competitive intramolecular silylation versus intermolecular silylation. Reaction conditions: **1** (0.1 mmol), **2** (0.2 mmol), [Rh(C$_2$H$_4$)$_2$Cl]$_2$ (5 mol%), **L3** (5 mol%), Et$_3$SiH (0.5 equiv.), toluene (1.0 mL) in a sealed Schlenk tube. Isolated yields were given. Enantiomer ratio determined by HPLC analysis on a chiral stationary phase. r.t. = room temperature.

methyl group on the phenyl ring. It was seen that the position of the methyl group did not undermine the reaction. We then examined a number of different groups para to the Si (**1n-1r**). Again, the electronic demands of these groups did not significantly affect the efficiency and enantioselectivity of the reaction. Thus, electron-withdrawing F- (**1n**), Cl- (**1o**), electron-donating MeO- (**1p**), CF$_3$O- (**1q**) and electron neutral Ph- (**1r**) substitutions all saw yields in 49–68% range and enantioselectivities around 90% ee. The absolutely configuration of product **3pn** was established by X-ray crystallography; (2) Frameworks other than the phenyl ethers were also successful. The 1-naphthalene (**1t**) and 1-ethylether-2-naphthalene (**1 s**) frameworks provided the corresponding products in good yields and high enantioselectivities (89–96% ee). Importantly, the acyclic stereogenic Si-H products are shown in Figs. 2–4 had never been reported. The structures of **3 h** and **3i** marked a fresh level of

complexity of stereogenic silicons available to date. The polysubstitution pattern shown in Fig. 4 (e.g., **3n** and **3o**) suggests that our products should be amenable to peripheral elaboration via standard reactions such as Suzuki couplings and S$_N$Ar reactions.

**Synthetic elaborations and control experiments**. To showcase the synthetic utility of our method, we first carried out a gram-scale synthesis of compound **3aa**, which smoothly afforded the desired product in good yield (60%) and excellent enantioselectivity (90% ee). We then explored the elaboration of the stereogenic Si–H in compound **3aa** under two set of conditions (Fig. 5a): (1) Pd/C catalyzed silane oxidation afforded silanol **3aa-1** in a 92% yield with 90% ee; (2) Hydrosilylation with 3-hexyne under Karstedt catalyst produced the vinylsilane **3aa-2** in a 93% yield with 89% ee. In both cases the enantio-purity of product was

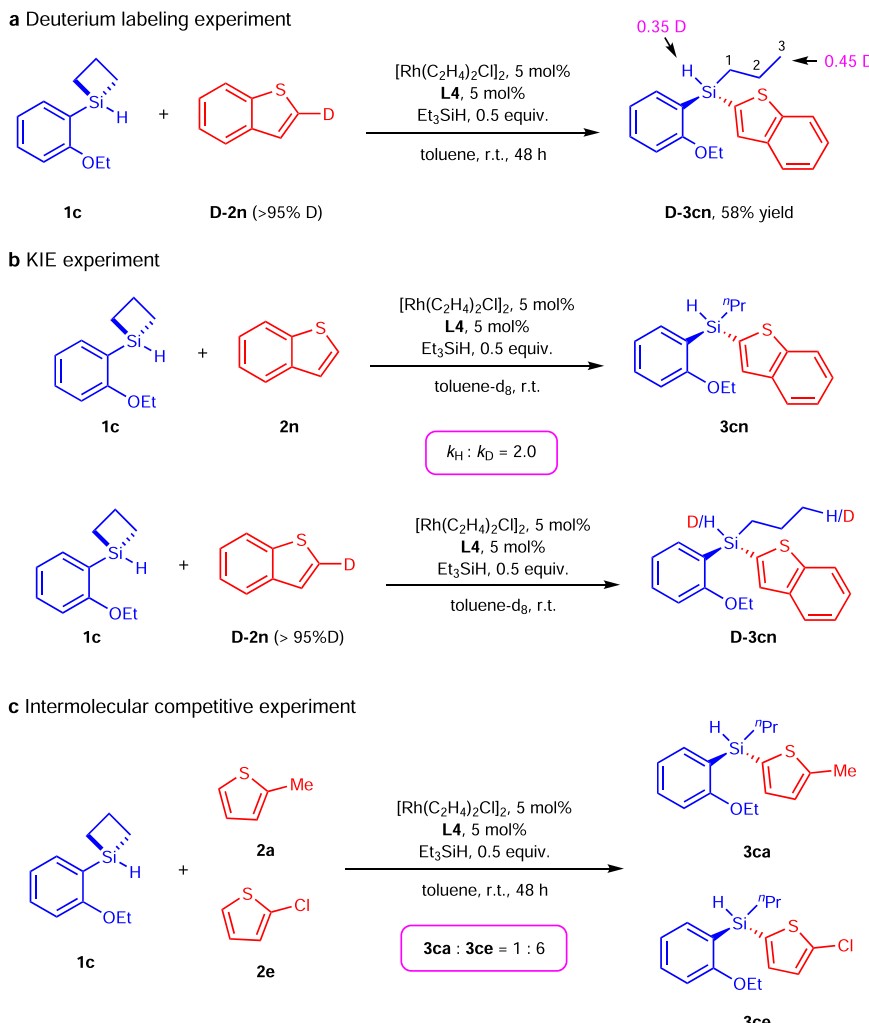

**Fig. 6 Mechanistic studies. a** Deuterium labeling experiment. Reaction conditions: **1c** (0.1 mmol), **D-2n** (0.1 mmol), [Rh(C₂H₄)₂Cl]₂ (5 mol%), **L4** (5 mol %), Et₃SiH (0.5 equiv.), toluene (1.0 mL) in a sealed Schlenk tube. Isolated yield was given. **b** Determination of kinetic isotope effect from two parallel reactions. **c** Intermolecular competition experiment of thiophenes. Reaction conditions: **1c** (0.1 mmol), **2a** (0.2 mmol), **2e** (0.2 mmol), [Rh(C₂H₄)₂Cl]₂ (5 mol%), **L4** (5 mol%), Et₃SiH (0.5 equiv.), toluene (1.0 mL) in a sealed Schlenk tube. The ratio of **3ca** to **3ce** was determined by $^1$H NMR analysis. r.t. = room temperature.

not compromised, suggesting that the Si–H was amenable to stereospecific transformations to access other stereogenic silicons. Besides, we also conducted control experiments for the direct construction of chiral tetraorganosilicons (Fig. 5b, c). No reaction took place in the case of methyl-substituted SCB **1u** with benzothiophene **2n** under standard conditions. When the reaction mixture was heated to 80 °C, only trace intermolecular silylation product **3un** could be detected by GC-MS analysis. Similar results were also obtained in the reaction of diphenylsilacyclobutane (Supplementary Fig. 13), which underscored the different reactivities between the current work with our previous report[67,68]. Significantly, when a possible intramolecular silylation site of SCB substrate was available, as opposed to the cases of **1h** and **1i** wherein the sites were blocked, intramolecular C–H silylation prevailed as seen in the cases of **1v** and **1w**. The tandem intramolecular C–H silylation/intermolecular dehydrogenative coupling products **3va~3wn** were consistent with our earlier work[54], albeit lower enantioselectivities were observed.

**Mechanism Investigation**. We then carried out deuterium labeling experiments to shed light on the reaction mechanism (Fig. 6). First, the reaction of SCB **1c** with deuterated benzothiophene **D-2n** proceeded smoothly under [Rh]-H catalysis to afford the desired product **D-3cn** in 58% yield. H/D scrambling from $^1$H and $^2$H NMR spectra showed that Si-H of **D-3cn** was partially deuterated (0.35 D), and control experiments suggested H/D exchange was facilitated by [Rh]-H catalyst outside of the catalytic cycle between **D-2n** and [Si]-H source (Fig. 7a, Supplementary Fig. 18–20). Deuteration merely on the terminal carbon atom of *n*-propyl group of **D-3cn** was also detected. These results indicated that the β-hydride elimination was unlikely to be operative in the process. Second, the relative ratio of the initial rates of two parallel reactions using **1c** with **2n**, **1c** with **D-2n**, respectively, was determined to be 2.0, indicating that C–H bond cleavage might be involved in the rate-determining step. In our earlier intramolecular C–H silylation[67,68], C–H bond activation was found to be energetically irreversible but not rate-limiting because the KIE of parallel reactions was determined to be 1.0. The fact that the C–H activation is involved in the rate-limiting step in this intermolecular C–H silylation underscores the vast difference between these two seemingly related reactions. Thirdly, a competition experiment of **2a** and **2e** was carried out and the corresponding products **3ca** and **3ce** were obtained in a 1:6 ratio. The faster reaction of

**a** Putative pathway for the generation of **D-3cn**

**b** Plausible mechanism for intermolecular C–H silylation

**Fig. 7 Proposed reaction mechanism. a** Putative pathway for the generation of **D-3cn**. **b** Plausible mechanism for intermolecular C–H silylation. [Rh*] = diphosphine coordinated rhodium.

thiophene bearing electron-withdrawing group suggested a faster oxidative addition process for the C–H bond cleavage, which further indicated C–H bond activation was involved in the rate-limiting step.

Based on the above results a plausible mechanism is depicted in Fig. 7b. The reaction initiates with the coordination of thiophene to the pre-formed [Rh]-H catalyst, which was identified as the catalyst resting state, followed by oxidative addition of silacyclo-butane (SCB) onto Rh(I), generating a five-membered rhodacycle **A**. This is the enantio-determining step based on our previous DFT calculations[68]. Reductive elimination occurs on Rh(III) cycle **A**, forming the Rh(I) intermediate **B** that is capable of C–H silylation. Subsequent C-H activation by Rh(I) **B** is perhaps the rate-limiting step. In a substrate that lacks steric hindrance around the Si atom, the Rh(I) intermediate **B** would react with the silane precursor, leading to oligomerization side reaction of the SCBs. However, in virtue of the steric shielding of the R group in Rh(I) intermediate **B**, C–H bond activation becomes competitive to give oxidative addition intermediate **C**. A final reductive elimination would produce the desired acyclic stereo-genic Si-H product **3**.

In conclusion, we have developed a protocol that enables enantioselective intermolecular C–H silylation reactions, produc-cing a wide scope of acyclic stereogenic silanes in high efficiency and excellent enantioselectivity. This work clearly indicates that how to match the reactivity of the C–H bonds with the reactivity

of the prochiral silanes is the most important question. In this study, it is achieved by placing a steric blocking group next to the Si center, and employing a highly reactive [Rh]-H catalyst. Future efforts should be paid to the expansion of the scope of both the silylation reagents and the C–H partners, such that more general prochiral silicons and C–H bonds could be applied to the access of almost all kinds of acyclic stereogenic Si–H compounds, which in turn can be elaborated into many kinds of stereogenic silicons.

## Methods

**General procedure for the preparation of [Rh]-H catalyst**. $[Rh(C_2H_4)_2Cl]_2$ (7.8 mg, 0.02 mmol, 1.0 equiv.), DTBM-MeO-biphep (23.0 mg, 0.02 mmol, 1.0 equiv.) and toluene (2.0 mL) were added into a sealed tube in $N_2$-flushed glove box. The reaction mixture was stirred at room temperature for 30 minutes. Then $Et_3SiH$ (32 μL, 0.2 mmol, 10.0 equiv.) was added in one portion and the stirring continued for 4 h at 50 °C to afford the stock solution of Rh(**L3**)-H which was used directly without further purification.

**General procedure for the synthesis of stereogenic monohydrosilane 3**. SCB substrate **1** (0.1 mmol), thiophene **2** (0.2 mmol) and Rh(**L3**)-H stock solution (0.5 mL) were added into a sealed tube equipped with magnetic stirring bar in $N_2$-flushed glove box, and the total volume of toluene solution was adjusted to be 1.0 mL. The tube was removed from glovebox and stirred at room temperature for 48 h. After that the reaction mixture was diluted with dichloromethane (2.0 mL), and the organic layer was concentrated under reduced pressure. The residue was purified by preparative TLC to afford the corresponding monohydrosilane **3**.

## Data availability

All data generated in this study are provided in the Supplementary Information/Source Data file. The X-ray crystallographic data used in this study are available in the Cambridge Crystallographic Data Center (CCDC) under accession code 2053377 (**3pn**) [www.ccdc.cam.ac.uk/data_request/cif].

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

## Acknowledgements

This work was supported by the National Natural Science Foundation of China (No. 21625104, W.H.), (No. 21901235, Q.Z.), (No. 21971133, W.H.) and the National Key Research and Development Program of China (No. 2017YFA0505203, W.H.).

## Author contributions

K.A. and W.M. contributed equally to this work. W.H. conceived of the project. Q.Z. designed the experiments. K.A., W.M., L.L., T.H., and G.G. performed the research. Q.Z. and W.H. wrote the manuscript.

## Competing interests

The authors declare no competing interests.
