## [Peer Review File · Nature Communications]

REVIEWER COMMENTS

Reviewer #1 (Remarks to the Author):

He, Zhang and co-workers report here a new method for the synthesis of acyclic stereogenic Si–H compounds by intermolecular C–H bond silylation. I basically agree with the author's design that a blocking group on the aryl substituent could effectively prevent the decomposition of the Si–H products. The manuscript is well-written and the substrate scope of the present reaction is reasonably broad. Considering the difficulty in construction of enantioenriched acyclic silicon-stereogenic silanes via intermolecular C–H silylation, this work is recommended to publish in nature communications after addressing the following points:

- (1) Can double C–H silylation on the two α -position of thiophene happen by changing the ratio of SCB and thiophene to 2:1?
- (2) Why 3-carboxylate-thiophene (2l) performed better than 2-carboxylate-thiophene (2f) in enantioselectivity, the authors should give some comments.
- (3) Compounds 3h and 3i bearing hindered ortho aryl substitution are compatible in this process. How about simple 2-substituted phenyl SCB? What is the result of the competition between intramolecular and intermolecular C–H silylation?
- (4) In their previous work (*Angew. Chem. Int. Ed.* 2016, 55, 6319), the substituted SCB (not Si–H) was used as an effective reagent. Can this reaction be conducted with the alkyl or aryl substituted SCB?
- (5) The author claimed "We also attempted SCBs bearing unsubstituted phenyl (i.e., 1a devoid of the -OBn group), but the reaction substrates suffered complete decomposition and no identified product was isolated." What are the main byproducts in the reaction?
- (6) How about other heterocycles in this process, such as furan, pyrrole?
- (7) For products, 3fa, 3ha, 3hi, 3ji, 3oa, 3oi and 3sa, the separation of two enantiomers peaks were not good. The authors should retry separation of two enantiomers and report accurate experimental results.

Reviewer #2 (Remarks to the Author):

Comments:

He and coworkers reported a study on an enantioselective intermolecular silylation between prochiral silanes and thiophenes. Intermolecular C-H silylation is an important method to construct the stereogenic Si-H compounds, however, as the authors note, this reaction usually suffers from various challenging problems, such as the decomposition of the silane substrates, low reactivity of intermolecular C-H bond and low stability of acyclic chiral silanes. In this manuscript, the authors overcame the problem by placing a steric blocking group next to the Si center and developing a highly reactive Rh-H catalyst. The study is inspiring and pioneering. But obviously, this work is not so perfect in some aspects, such as substrate scope and mechanistic study. For example, although the authors have finished 71 examples, most of which are similar in structure type. And the substrate scope of the heterocyclic arenes only include thiophenes. Moreover, the effect of the ortho-substituents on the aryl silane substrates makes the reaction impractical in most cases. That is to say, this reaction may not be as good as the authors describe. Considering the innovation of the reaction design and the integrity of research work, I would recommend publication of the manuscript in Nature Communications after major revisions:

- 1) Fig.1c is misleading. The representative C-H partners in this manuscript only include (benzo)thiophenes, but the authors conclude them as general heterocyclic arenes. Are other heterocycles (e.g. pyridine, indole

and furan) reactive in this reaction? If not, the description of "Het" is inappropriate.

2) The unsubstituted phenyl silanes (fig. 2) and compound 1r (fig. 4) suffer from the complete decomposition, what's the decomposed product?

3) The enantiodetermining step (fig. 5) is the oxidative addition of SCB to generate rhodacycle Rh(III) A, this step seems to be controlled by the difference between ortho-substituted phenyl and hydrogen atom. However, according to the experiment data, it is interesting that steric hindrance of ortho substituents does not seem to have much effect on the enantioselectivities (fig. 3). I suggest that authors add some examples of other ortho-substitution to assist the understanding.

4) In most cases, the yields are in 50-70% range, how about the resting 30-50%? Are the substrates not completely converted or is there a side reaction?

5) Why is thiophene necessary in this transformation? I am curious to know whether the cooperation of "S" atom on thiophene can lead to an activation of C-H bond. What's the resting state of this catalytic reaction?

6) 5 mol% is a little high for a Rh-catalyzed reaction. What happens when a lower catalyst loading is used?

7) I cannot find the spectra for D-3cn in SI, more detailed characterization data are necessary, besides, the hydrogen deuterium ratio also help explain the mechanism. Additionally, the concentration of 3cn and D-3cn is high (~0.01 M) at 0 min, what happens at the beginning of the reaction?

Some typos:

Line 135, "2-methylthiopene" should be "2-methylthiophene".

Line 139, "enantioselectivities" should be "enantioselectivities".

Reviewer #3 (Remarks to the Author):

Silacyclobutanes (SCBs) have been studied for a long time for their reactivity derived from their unique ring strain. These reactions are proposed to involve five-membered metallocycle intermediates via oxidative insertion into the SCBs, followed by the migratory insertion of the unsaturated carbon-carbon bonds or intramolecular C-H activation to form cyclic silane derivatives. Despite of these interesting transformations, enantioselective intermolecular C-H silylation based on SCBs remains unknown to date. He and co-workers reported an enantioselective intermolecular C-H silylation of SCBs with thiophenes by using rhodium hydride catalyst, affording various chiral organosilanes containing silicon-stereogenic center. The various functional group could be tolerated, and the average ee value is around 88%. The construction of silicon-stereogenic center is extremely challenging due to the instability of sp² hybrid silicon and the tendency of racemization of silicon-stereogenic center via Si(V). In addition, the author showcase the synthetic utility through gram-scale synthesis and the elaboration of the stereogenic Si-H. Some control experiments were also conducted to shed light on the reaction mechanism. The authors propose that the oxidative addition of SCB onto Rh(I), generating the five-membered rhodacycle A, is the enantio-determining step of this reaction, i.e. the silicon stereocenter was constructed through the desymmetrization of C-Si bond activation in SCBs. Due to the challenge on chiral silicon-stereogenic center, this manuscript could be considered after following revision.

(1) In Fig.2-4, the desired chiral silanes products were obtained in about 60% yields in most cases, the moderate yield was due to the unreacted starting materials or other by-products? Also, this suggests an opportunity for improvement.

(2) For the synthesis of enantioenriched silicon-stereogenic organosilanes, the literatures on other attractive desymmetrization of Si-C bonds should be cited, such as J. Am. Chem. Soc. 2015, 137, 11838, Angew. Chem. Int. Ed. 2020, 59, 11927, and Angew. Chem. Int. Ed. 2020, 59, 790.

(3) The Ligand in Table 1 should be bold type, so does the footnote in the Fig 1-4.

(4) I noticed that in this intriguing enantioselective intermolecular C–H silylation of silacyclobutanes, the author showcased the scope of C–H bond partners only in the forms of thiophenes and benzothiophene. Here is the question: how about other small heterocycle framework, or simple arene?

(5) As the author assumed, the ortho-substituted substrate (i. e. blocking group) likely prevent the [Rh]-Si self-reaction by virtue of steric hindrance. However, for the ortho-aryl substituted SCBs substrates, I think it's easier to undergo intramolecular C–H silylation than intermolecular reaction with thiophenes (cited in ref 71 and Org. Lett. 2015, 17, 386). Compared to the works before, how about the asymmetric intramolecular reactions under [Rh]-H catalysis?

(6) The relative ratio of the initial rates of two parallel reactions using 1c with 2n, 1c with D-2n, respectively, indicated that the C–H activation is involved in the rate-limiting step. Here is a question, in the deuterium-labeling experiment with D-2n, was the position of the deuterium atom introduced definite? Intermediate A may undergo reversible β -H elimination followed by reductive elimination, which may result in deuterium incorporation at both C1 and C2 positions of the n-propyl group.

(7) The key rhodium hydride catalyst Rh(L5)-H was synthesized according previous reports. It's purity and enantiopurity needs to be included in SI, and compared with previous analysis.

(8) Some of integration of HPLC are not reasonable because the two retention peaks were not separated at all, such as compound 3ai, 3ha, 3hi, 3ji, 3ma, 3mn, 3oa, 3oi, 3sa, 3si, 3sn, 3ti, and 3tn. On the other hand, in some HPLC, such 3kn, 3li, 3na, 3pa, 3ri, it looks that small impurity has been involved the integration obviously, which makes me doubt the accuracy of the enantioselectivity values of the chiral compounds and yields given by the author. This problem in HPLC spectrograms combined with the corresponding NMR spectrum provided, the purities of the corresponding compounds should reconsidered.

Reviewer #1

He, Zhang and co-workers report here a new method for the synthesis of acyclic stereogenic Si–H compounds by intermolecular C–H bond silylation. I basically agree with the author's design that a blocking group on the aryl substituent could effectively prevent the decomposition of the Si–H products. The manuscript is well-written and the substrate scope of the present reaction is reasonably broad. Considering the difficulty in construction of enantioenriched acyclic silicon-stereogenic silanes via intermolecular C–H silylation, this work is recommended to publish in nature communications after addressing the following points:

Reply: We appreciate your positive comments. Your kind suggestions also help us a great deal in improving our manuscript. Please see below for our detailed responses.

(1) Can double C–H silylation on the two α -position of thiophene happen by changing the ratio of SCB and thiophene to 2:1?

Reply:

Thank you for your question. At your request we conducted the experiment using 1.0 equivalent of thiophene **2h** and 2.0 equivalent of SCB **1a** under standard conditions. The tertiary hydrosilane **3ah** was isolated in 57% yield, yet the target bis-silylation product was not detected. The excess SCB substrate was fully converted into oligomerization byproduct with unreacted thiophene remained. We reckoned that there existed a competition between the self-reaction of SCB and the current C–H activation of thiophene once Si–[Rh] intermediate was formed, therefore the excess addition of thiophene partner was crucial towards the desired transformation.

(2) Why 3-carboxylate-thiophene (**2l**) performed better than 2-carboxylate-thiophene (**2f**) in enantioselectivity, the authors should give some comments.

Reply: Thank you for your question. We have identified the thiophene coordinated rhodium hydride species to be the catalyst resting state (see Supporting Information for details), since

it might affect both enantioselectivity and reactivity of the oxidative addition process of SCB onto Rh^I. For thiophene partner **2f**, both sulfur atom and adjacent oxygen atom of carbonyl group would bond to the rhodium center, giving rise to bidentate coordinated complex. Oxidative addition of Si-C bond onto such rhodium complex might affect enantioselectivity as this process was considered to be enantio-determining based on our previous DFT calculations. For thiophene partner **2i**, the carbonyl group was placed away from the coordinated rhodium center, therefore better enantioselectivity was observed.

(3) Compounds **3h** and **3i** bearing hindered ortho aryl substitution are compatible in this process. How about simple 2-substituted phenyl SCB? What is the result of the competition between intramolecular and intermolecular C–H silylation?

Reply:

3va, 86%, 31% ee
3vi, 87%, 56% ee
3vn, 87%, 46% ee

3wa, 82%, 75% ee
3wi, 84%, 61% ee
3wn, 85%, 86% ee

Thank you for your question. Per your suggestion we surveyed the corresponding substrates with their possible intramolecular C-H activation sites available. As seen in the cases of **1v** and **1w**, the intramolecular C-H activation prevailed and the competitive intermolecular silylation products were not detected. The tandem intramolecular C-H silylation/intermolecular dehydrogenative coupling products **3va-3wn** were consistent with our earlier work (ref. 53), albeit lower enantioselectivities were observed.

(4) In their previous work (*Angew. Chem. Int. Ed.* 2016, 55, 6319), the substituted SCB (not Si–H) was used as an effective reagent. Can this reaction be conducted with the alkyl or aryl

substituted SCB?

Reply:

Thank you for your question. As suggested we monitored the reaction of methyl-substituted SCB **1u** with benzothiophene **2n** under standard conditions. No reaction took place at room temperature with both starting materials fully recovered. When the reaction temperature was increased to 80 °C, trace intermolecular silylation product **3un** could be detected by GC-MS analysis. Similar results were also observed in the case of diphenylsilacyclobutane under the same conditions (see Supporting Information for details), which underscored the different reactivities of the current work with our previous report.

(5) *The author claimed "We also attempted SCBs bearing unsubstituted phenyl (i.e., 1a devoid of the -OBn group), but the reaction substrates suffered complete decomposition and no identified product was isolated." What are the main byproducts in the reaction?*

Reply: Thank you for pointing out such important issue. The SCB substrates without ortho-substitution were fully converted into oligomerization byproducts while the thiophenes remained intact. That is the reason we placed a blocking group in order to suppress the self-reaction of SCB by virtue of its steric hindrance.

(6) *How about other heterocycles in this process, such as furan, pyrrole?*

Reply: Thank you for pointing out this important issue. We tested 2-methylfuran, *N*-phenylpyrrole, *N*-methylindole, and quinoline as potential arene partners, but they were not compatible in current reactions and no desired silylation products were detected. The SCB substrate gradually oligomerized while the heterocyclic partners remained intact. We reckoned that the initial coordination of thiophenes to the rhodium center made the formal intermolecular silylation into the intramolecular one, thus overcoming the low reactivities of C-H bonds. But for other heterocycles, the coordination between soft acid (rhodium) and hard bases (nitrogen or oxygen) didn't match, which resulted in the prevalence of self-reaction of SCB substrate.

(7) For products, **3fa**, **3ha**, **3hi**, **3ji**, **3oa**, **3oi** and **3sa**, the separation of two enantiomers peaks were not good. The authors should retry separation of two enantiomers and report accurate experimental results.

Reply: Thank you for your kind advice. We have revised the separation conditions of the enantioenriched samples of **3fa**, **3ha**, **3hi**, **3ji**, **3oa**, **3oi** and **3sa**. The updated data have been included in the main text as well as the Supporting Information.

Reviewer #2

Comments:

He and coworkers reported a study on an enantioselective intermolecular silylation between prochiral silanes and thiophenes. Intermolecular C-H silylation is an important method to construct the stereogenic Si-H compounds, however, as the authors note, this reaction usually suffers from various challenging problems, such as the decomposition of the silane substrates, low reactivity of intermolecular C-H bond and low stability of acyclic chiral silanes. In this manuscript, the authors overcame the problem by placing a steric blocking group next to the Si center and developing a highly reactive Rh-H catalyst. The study is inspiring and pioneering. But obviously, this work is not so perfect in some aspects, such as substrate scope and mechanistic study. For example, although the authors have finished 71 examples, most of which are similar in structure type. And the substrate scope of the heterocyclic arenes only include thiophenes. Moreover, the effect of the ortho-substituents on the aryl silane substrates makes the reaction impractical in most cases. That is to say, this reaction may not be as good as the authors describe. Considering the innovation of the reaction design and the integrity of research work, I would recommend publication of the manuscript in Nature Communications after major revisions:

Reply: Thank you for your objective summary and favorable remarks on our manuscript. We have also carried out additional work to address your insightful questions. Please see below:

1) Fig.1c is misleading. The representative C-H partners in this manuscript only include (benzo)thiophenes, but the authors conclude them as general heterocyclic arenes. Are other heterocycles (e.g. pyridine, indole and furan) reactive in this reaction? If not, the description of "Het" is inappropriate.

Reply:

c) This Work: Enantioselective Intermolecular C-H Silylation

Thank you for pointing out this important issue. We tested pyridine, *N*-methylindole and 2-methylfuran as potential arene partners. Unfortunately those heterocycles were not compatible in current reactions and no desired silylation products were detected. To this end we have revised the structure shown in Figure 1c to avoid potential misleading.

2) The unsubstituted phenyl silanes (fig. 2) and compound **1r** (fig. 4) suffer from the complete decomposition, what's the decomposed product?

Reply: Thank you for your question. For SCBs without ortho-substitution, the substrates rapidly oligomerized while the thiophenes remained intact. For compound **1r**, the substrate was not compatible with [Rh]-H catalysis and was fully converted into self-reaction byproduct, therefore the less reactive [Rh]-Cl catalyst was employed to furnish the desired products.

3) The enantiodetermining step (fig. 5) is the oxidative addition of SCB to generate rhodacycle Rh(III) A, this step seems to be controlled by the difference between ortho-substituted phenyl and hydrogen atom. However, according to the experiment data, it is interesting that steric hindrance of ortho substituents does not seem to have much effect on the enantioselectivities (fig. 3). I suggest that authors add some examples of other ortho-substitution to assist the understanding.

Reply: Thank you for your insightful suggestions. You were right on the point that the enantioselectivity was determined by the steric difference between *H* atom and ortho-substituted aryl group. In essentials the ortho-substituent on the aryl ring functioned as a blocking group to suppress the self-reaction of SCB. Since the steric difference between the smaller *H* atom and the larger aryl ring was distinct enough, it was not surprising that the steric hindrance of ortho-substituent had less effect on the enantioselectivities. Previously we had attempted intramolecular desymmetrization of methyl-substituted silacyclobutane (see ref. 53 for details), but almost no enantioselectivity was observed, which was probably attributed to the close steric hindrance between methyl and ortho-substituted aryl group.

4) In most cases, the yields are in 50-70% range, how about the resting 30-50%? Are the substrates not completely converted or is there a side reaction?

Reply: Thank you for your question. The SCB substrates were fully converted and the competitive oligomerization of strained SCB accounted for the mass balance. For illustration purpose we took the reaction of SCB **1c** with *N*-methylindole under Rh(**L4**)-H catalysis as an example.

At first we employed ^1H NMR spectroscopy to monitor the reaction process. After 24 hours the SCB substrate was fully converted while *N*-methylindole remained intact as depicted in Figure R1. Since there was no characteristic peak which could be assigned to any intermediates or byproducts from the crude ^1H NMR spectrum, the reaction mixture was diluted with dichloromethane and concentrated under vacuum. The residue was passed through a short pad of silica gel using THF as eluent and subjected to GPC analysis. As shown in Figure R2, the small M_n of 668 indicated the byproduct was a mixture of oligomers derived from self-reaction of strained SCB substrate. Furthermore, we also pursued LC-MS analysis for its higher sensitivity. As depicted in Figure R3 & R4, a characteristic m/z value of 383 (APCI), 577 (APCI) and 790 (ESI) could be detected, which was in good agreement with the structure of dimer ($[\text{M}-\text{H}]$), trimer ($[\text{M}+\text{H}]$) and tetramer ($[\text{M}+\text{Na}]$) of SCB monomer, respectively.

Figure R1. ^1H NMR monitoring on the reaction of SCB **1c** with *N*-methylindole

	[min]	[mV]	[mol]	Mn	668
Peak start	9.587	3.614	5,563	Mw	1,505
Peak top	10.462	26.521	945	Mz	2,282
Peak end	11.018	16.333	39	Mz+1	2,856
				Mv	1,505
Height [mV]			15.133	Mp	1,900
Area [mV*sec]			673.664	Mz/Mw	1.516
Area% [%]			100.000	Mw/Mn	2.252
[eta]			1505.17926	Mz+1/Mw	1.897

Figure R2. GPC spectrum of the self-reaction byproduct of SCB **1c**

Figure R3. LC-MS (APCI) spectrum of the self-reaction byproduct of SCB **1c**

Figure R4. LC-MS (ESI) spectrum of the self-reaction byproduct of SCB **1c**

5) Why is thiophene necessary in this transformation? I am curious to know whether the cooperation of "S" atom on thiophene can lead to an activation of C-H bond. What's the resting state of this catalytic reaction?

Reply: Thank you for your question. To identify the resting state of the catalytic cycle, we surveyed the reaction of SCB **1c** with benzothiophene **2n** under Rh(**L4**)-H catalysis step by step with NMR monitoring. After the Rh(**L4**)-H stock solution was prepared *in situ*, benzothiophene **2n** was added and the signal of ^{31}P NMR shifted to higher magnetic fields from 35.5 ppm to 31.9 ppm which was probably attributed to the coordination of benzothiophene to rhodium center. Then SCB **1c** was added into the reaction mixture and product **3cn** could be detected after the first ^1H NMR data acquisition. The thiophene coordinated rhodium hydride species was considered to be the resting state, as it might affect both enantioselectivity and reactivity of the oxidative insertion of Rh^{I} into SCB. For other heterocycles, the coordination between soft acid (rhodium) and hard bases (nitrogen or oxygen) didn't match. This also help to explain why only thiophenes were successfully in this reaction, because the initial coordination of thiophenes to the rhodium center made the formal intermolecular silylation into the intramolecular one, thus overcoming the low reactivities of C-H bonds.

6) 5 mol% is a little high for a Rh-catalyzed reaction. What happens when a lower catalyst loading is used?

Reply: Thank you for your insightful question. As suggested we conducted the template reaction in Table 1 using halved catalyst loading of 2.5 mol%. Under standard condition the tertiary hydrosilane **3aa** was obtained in 46% yield, with both starting materials remained and oligomerization side product detected. As the reactive $[\text{Rh}]\text{-H}$ catalyst was not quantitatively generated from the $[\text{Rh}]\text{-Cl}$ precursor (see Supporting Information for details), we thus employed 5 mol% of rhodium catalyst to ensure sufficient catalytic efficiency.

7) I cannot find the spectra for D-3cn in SI, more detailed characterization data are necessary, besides, the hydrogen deuterium ratio also help explain the mechanism. Additionally, the concentration of 3cn and D-3cn is high (~0.01 M) at 0 min, what happens at the beginning of the reaction?

Reply:

a) Deuterium Labeling Experiment

b) Control Experiments

c) Generation of D-3cn

Thank you for pointing out this important issue. The detailed characterization of **D-3cn** has been included in the Supporting Information. Besides, we also checked the H/D ratio of **D-3cn**

in the hope to give an in-depth answer. H/D scrambling from ^1H and ^2H NMR spectra showed that Si-H of **D-3cn** was partially deuterated (0.35 D), and deuteration on the terminal carbon atom of *n*-propyl group was also detected. Control experiments indicated the H/D scrambling was facilitated by [Rh]-H catalyst outside of the catalytic cycle between deuterated benzothiophene **D-2n** and [Si]-H source (see Supporting Information for details). Thirdly, we assigned the first NMR data acquiring as the beginning of timing, but it took us approximately 0.5 h for sample preparation in glove box and subsequent NMR analysis. The reaction proceeded smoothly at room temperature upon reagent mixing, and only both starting material and minor silylation product were detected from the first NMR spectrum.

Some typos:

Line 135, "2-methylthiopene" should be "2-methylthiophene".

Line 139, "enantioselectivies" should be "enantioselectivities".

Reply: Thank you for pointing out our clerical errors carefully. We have revised them in the main text.

Reviewer #3

*Silacyclobutanes (SCBs) have been studied for a long time for their reactivity derived from their unique ring strain. These reactions are proposed to involve five-membered metallocycle intermediates via oxidative insertion into the SCBs, followed by the migratory insertion of the unsaturated carbon-carbon bonds or intramolecular C-H activation to form cyclic silane derivatives. Despite of these interesting transformations, enantioselective intermolecular C-H silylation based on SCBs remains unknown to date. He and co-workers reported an enantioselective intermolecular C-H silylation of SCBs with thiophenes by using rhodium hydride catalyst, affording various chiral organosilanes containing silicon-stereogenic center. The various functional group could be tolerated, and the average ee value is around 88%. The construction of silicon-stereogenic center is extremely challenging due to the instability of sp^2 hybrid silicon and the tendency of racemization of silicon-stereogenic center via Si(V) . In addition, the author showcase the synthetic utility through gram-scale synthesis and the elaboration of the stereogenic Si-H. Some control experiments were also conducted to shed light on the reaction mechanism. The authors propose that the oxidative addition of SCB onto Rh(I) , generating the five-membered rhodacycle **A**, is the enantio-determining step of this reaction, i.e. the silicon stereocenter was constructed through the desymmetrization of C-Si bond activation in SCBs. Due to the challenge on chiral silicon-stereogenic center, this*

manuscript could be considered after following revision.

Reply: Thank you for your nice summary and encouraging comments on our manuscript. We have also carried out additional work to address your insightful questions. Please see below:

(1) In Fig.2-4, the desired chiral silanes products were obtained in about 60% yields in most cases, the moderate yield was due to the unreacted starting materials or other by-products ? Also, this suggests an opportunity for improvement.

Reply: Thank you for your question. The SCB substrates were fully converted and the competitive oligomerization of strained SCB accounted for the mass balance. As another referee also raised a similar question, please refer to our response to Reviewer 2, Question 4 for further details.

(2) For the synthesis of enantioenriched silicon-stereogenic organosilanes, the literatures on other attractive desymmetrization of Si–C bonds should be cited, such as J. Am. Chem. Soc. 2015, 137, 11838, Angew. Chem. Int. Ed. 2020, 59, 11927, and Angew. Chem. Int. Ed. 2020, 59, 790.

Reply: Accordingly, we have added these citations into the **References** section. Thank you for your kind reminder.

(3) The Ligand in Table 1 should be bold type, so does the footnote in the Fig 1-4.

Reply: We appreciate your helpful suggestion. We have revised them accordingly.

(4) I noticed that in this intriguing enantioselective intermolecular C–H silylation of silacyclobutanes, the author showcased the scope of C–H bond partners only in the forms of thiophenes and benzothiophene. Here is the question: how about other small heterocycle framework, or simple arene ?

Reply: Thank you for pointing out this important issue. We tested 2-methylfuran, pyridine, *N*-methylindole and *N*-phenylpyrrole as potential arene partners. Unfortunately those heterocycles were not compatible in current reactions and no silylation products were detected. Benzene and 1,3-difluorobenzene appeared to be unsuccessful either even under neat conditions.

(5) As the author assumed, the ortho-substituted substrate (i. e. blocking group) likely prevent the [Rh]-Si self-reaction by virtue of steric hindrance. However, for the ortho-aryl substituted SCBs substrates, I think it's easier to undergo intramolecular C–H silylation than intermolecular reaction with thiophenes (cited in ref 71 and *Org. Lett.* 2015, 17, 386). Compared to the works before, how about the asymmetric intramolecular reactions under [Rh]-H catalysis?

Reply: Thank you for your question. Per your suggestion we surveyed the corresponding substrates with their possible intramolecular C-H activation sites available. The intramolecular C-H activation prevailed and the competitive intermolecular silylation products were not detected. As another referee also raised a similar question, please refer to our response to Reviewer 1, Question 3 for further details.

(6) The relative ratio of the initial rates of two parallel reactions using 1c with 2n, 1c with D-2n, respectively, indicated that the C–H activation is involved in the rate-limiting step. Here is a question, in the deuterium-labeling experiment with D-2n, was the position of the deuterium atom introduced definite? Intermediate A may undergo reversible β -H elimination followed by reductive elimination, which may result in deuterium incorporation at both C1 and C2 positions of the n-propyl group.

Reply: Thank you for pointing out this important issue. At your request we conducted deuterium labelling experiment using SCB **1c** and deuterated benzothiophene **D-2n** in the hope to give an in-depth answer. H/D scrambling from ^1H and ^2H NMR spectra showed that Si-H of **D-3cn** was partially deuterated, and deuteration merely on the terminal carbon atom of n-propyl group was also detected. These results indicated that the β -hydride elimination was unlikely to be operative in the process. Control experiments indicated the H/D scrambling was facilitated by [Rh]-H catalyst outside of the catalytic cycle between deuterated benzothiophene **D-2n** and [Si]-H source. As another referee also raised a similar question, please refer to our response to Reviewer 2, Question 7 for further details.

(7) The key rhodium hydride catalyst Rh(L5)-H was synthesized according previous reports. It's purity and enantiopurity needs to be included in SI, and compared with previous analysis.

Reply: Thank you for your helpful suggestion. Such information is not only necessary for the curious minds, but also valuable for future studies. Thus, the yields of Rh(**L3**)-H and Rh(**L4**)-

H were included in the Supporting Information accordingly. Besides, unlike Rh(**L5**)-H in our previous report, the key rhodium hydride species Rh(**L3**)-H was not generated quantitatively from [Rh]-Cl precursor. We attempted numerous trials to isolate Rh(**L3**)-H in pure form to test its enantiopurity, but the hydride catalyst was only stable in solution, and recrystallizing was also troubled by its great solubility. Therefore the [Rh]-H stock solution was used directly without further purification and we revised the reaction conditions described in Table1 and Figure 1-7 to avoid potential misleading.

(8) Some of integration of HPLC are not reasonable because the two retention peaks were not separated at all, such as compound 3ai, 3ha, 3hi, 3ji, 3ma, 3mn, 3oa, 3oi, 3sa, 3si, 3sn, 3ti, and 3tn. On the other hand, in some HPLC, such 3kn, 3li, 3na, 3pa, 3ri, it looks that small impurity has been involved the integration obviously, which makes me doubt the accuracy of the enantioselectivity values of the chiral compounds and yields given by the author. This problem in HPLC spectrograms combined with the corresponding NMR spectrum provided, the purities of the corresponding compounds should reconsidered.

Reply: Thank you for your kind advice. We have revised the separation conditions of the enantioenriched samples listed above. The updated data have been included in the main text as well as the Supporting Information.

REVIEWERS' COMMENTS

Reviewer #1 (Remarks to the Author):

On the basis of the comments from the three Reviewers, the authors have addressed the main suggestions and improved the manuscript to a stage that I feel is suitable for publication in Nature Communications.

Reviewer #2 (Remarks to the Author):

He, Zhang and coworkers have largely addressed my concerns and I support its publication after addressing this following point.

Please check to make sure the key and relevant references have cite appropriately (for example, Fang et al. JACS, 2017, 139, 11601-11609)

Reviewer #3 (Remarks to the Author):

The manuscript had been well revised. It can be published at present form.

Reviewer #1

On the basis of the comments from the three Reviewers, the authors have addressed the main suggestions and improved the manuscript to a stage that I feel is suitable for publication in Nature Communications.

Reply: We appreciate your positive comments. Your kind suggestions also help us a great deal in improving our manuscript.

Reviewer #2

He, Zhang and coworkers have largely addressed my concerns and I support its publication after addressing this following point.

Reply: Thank you for your favorable remarks. We have revised the manuscript accordingly to address your remaining concern.

Please check to make sure the key and relevant references have cite appropriately (for example, Fang et al. JACS, 2017, 139, 11601-11609)

Reply: Thank you for your kind reminder. Per your kind suggestion, this paper has been cited as ref. 37 in the revised manuscript.

Reviewer #3

The manuscript had been well revised. It can be published at present form.

Reply: We are grateful for your encouraging remarks. We have benefited a lot from your insightful suggestions to improve our manuscript.